# A Rare Case of Tricuspid Valve Libman–Sacks Endocarditis in a Pregnant Woman with Primary Antiphospholipid Syndrome

**DOI:** 10.3390/jcm11195875

**Published:** 2022-10-05

**Authors:** Sonia Migliorini, Ciro Santoro, Alessandra Scatteia, Santo Dellegrottaglie, Antonella Tufano, Vittoria Cuomo, Emanuele Pilato, Giuseppe Comentale, Maria D’Armiento, Maurizio Guida, Laura Sarno

**Affiliations:** 1Department of Neuroscience, Reproductive Sciences and Dentistry, School of Medicine, University of Naples Federico II, 80131 Naples, NA, Italy; 2Department of Advanced Biomedical Sciences, Federico II University Hospital, 80131 Naples, NA, Italy; 3Division of Cardiology, “Villa dei Fiori” Hospital, 80131 Naples, NA, Italy; 4Zena and Michael A. Wiener, Cardiovascular Institute/Marie-Josee and Henry R. Kravis Center for Cardiovascular Health, Icahn School of Medicine at Mount Sinai, New York, NY 10029, USA; 5Department of Clinical Medicine and Surgery, Federico II University Hospital, 80131 Naples, NA, Italy; 6Department of Advanced Biomedical Science, Division of Adult and Pediatric Cardiac Surgery, University of Naples Federico II, 80131 Naples, NA, Italy; 7Pathology Unit, Department of Public Health, School of Medicine, University of Naples Federico II, 80131 Naples, NA, Italy

**Keywords:** Libman–Sacks endocarditis, pregnancy, thrombosis, primary antibody syndrome, antiphospholipid syndrome

## Abstract

Antiphospholipid Antibody Syndrome (APS) is a systemic autoimmune disease characterized by acquired hypercoagulability with the possible development of venous, arterial, and microvascular thrombosis. We report a rare case of Libman–Sacks tricuspid valve endocarditis in a 38-year-old pregnant woman at 15 weeks gestation with unknown primary antiphospholipid syndrome. During a routine cardiac examination and echocardiography performed for a previous episode of pleuropericarditis, a large, mobile mass with irregular edges was found at the level of the tricuspid valve. Three main differential diagnoses for intramyocardial mass were examined: tumor, infective endocarditis, and nonbacterial thrombotic endocarditis (NTBE). Cardiac magnetic resonance imaging (CMR) with contrast raised the suspicion of a thrombus. The woman was hospitalized urgently at the Cardiac Intensive Care Unit of the Federico II University Hospital, and anticoagulant and antiplatelet therapy were started. The thrombophilic screening performed and medical history confirmed the diagnosis of primary antibody syndrome (APS). A multidisciplinary consultation with obstetricians, cardiologists, anesthetists, and cardiac surgeons was required. The patient decided not to terminate the pregnancy despite the risk to her health and to undergo cardiac surgery during pregnancy. Histological examination confirmed the presence of nonbacterial thrombotic endocarditis. Weekly obstetric scans were performed after surgery to verify fetal well-being. An emergency cesarean section was performed at the 35th week of gestation due to repeated deceleration and abnormal short-term variability on c-CTG in a pregnancy complicated by fetal growth restriction and gestational hypertension. A newborn weighing 1290 g was born. She was hospitalized in Neonatal Intensive Care and discharged after two months; currently, she enjoys good health. The management of patients with antiphospholipid antibody syndrome has not yet been standardized, but there is a general consensus that patients who do not have thrombocytopenia, thromboembolic phenomena, or pregnancy should not undergo any treatment or should take only low doses of acetylsalicylic acid. In the presence of any of the above conditions, various treatment regimens have been used based on the severity and individuality of the case.

## 1. Introduction

The antiphospholipid syndrome (APS) is a systemic autoimmune disorder of acquired hypercoagulability characterized by venous, arterial, and microvascular thrombosis and increased pregnancy morbidity in the presence of persistently positive antiphospholipid antibodies and constitutes a major cause of cardiovascular events in young people. Cardiac involvement in APS may present as heart valve disease, affecting approximately a third of patients, or, less frequently, as intracardial thrombosis, nonbacterial thrombotic endocarditis (i.e., Libman–Sacks endocarditis), pulmonary hypertension, right or left ventricular dysfunction, micro-vascular thrombosis, coronary artery, or micro-vascular disease with overt or silent clinical presentation. Libman–Sacks endocarditis is a type of sterile nonbacterial thrombotic endocarditis (NBTE) secondary to inflammation. This term describes vegetations on the cardiac valves that are sterile and do not show any signs of infection. Libman–Sacks endocarditis most commonly affects the mitral followed by aortic valves, but other valves may also be involved. The initial development of Libman–Sacks endocarditis appears to be an endothelial injury in the setting of a hypercoagulable state. So, they are mainly observed in patients with malignancies (mainly solid tumors; adenocarcinoma); systemic lupus erythematosus (SLE), which was first described in women in 1985; and antiphospholipid antibody syndrome (APS) [1]. Patients with APS have endothelial dysfunction, accelerated proliferation and hyperplasia, atherogenesis, platelet activation and aggregation, inflammatory product secretion, and coagulation-fibrinolytic dysregulation. APS is diagnosed when a patient meets at least one clinical and one laboratory criterion. The clinical criteria are (1) one or more clinical episodes of arterial, venous, or small vessel thrombosis in any tissue or organ confirmed by imaging or histopathology, or (2) pregnancy morbidity. The laboratory criteria, all of which involve the presence of antiphospholipid (aPL) antibody, are (1) positive lupus anticoagulant (LA) antibody, (2) moderate to high levels of isotype IgG or IgM anticardiolipin (aCL) antibody, or (3) isotype IgG or IgM anti-beta-2 glycoprotein I (anti-β2GPI) antibody on at least two occasions at least 12 weeks apart [2,3,4,5,6,7,8]. APS is the most important autoimmune disease responsible for severe complications in pregnancy, together with *systemic lupus erythematosus* [9]. Indeed, pregnancy is a pro-thrombotic condition that can promote thrombotic events in women with acquired thrombophilia. Moreover, APS has been reported to be associated with a high risk of feto-maternal complications, such as stillbirths, intrauterine demise, preterm birth, preeclampsia, placenta abruption, and fetal growth restriction. Despite low-molecular-weight heparin (LMWH) and low-dose aspirin having been established as the standard of care for pregnant women with APS, up to 30% of pregnant women with APS continue to have complications [10]. We report a case of unknown APS in a pregnant woman presenting with Libman–Sacks endocarditis affecting the tricuspid sub-valvular apparatus. 

## 2. Case Report

A 38-year-old pregnant woman, at 15 weeks of gestation, was referred to the High-Risk Pregnancy Clinic, Mother and Child Department, University Hospital Federico II, in Naples, Italy. She was in her sixth pregnancy, reporting two previous stillbirths and three previous pregnancies, all complicated by fetal growth restriction. Of note, the last miscarriage occurred along with pleuropericarditis. Since then, she remained asymptomatic and had no limitations to exercise tolerance. She did not report any chronic disease and her personal and family history were negative for thromboembolic events. According to local protocols, a cardiologic examination was requested. Cardiac and lung examinations, as well as the electrocardiogram, showed no significant findings. The patient was apyretic, cuff-blood pressure was 110/60 mmHg, heart rate was 80 beats per minute, and oxygen saturation was 97% in room air. Transthoracic echocardiography revealed a large mobile mass with irregular borders, attached to the sub-valvular apparatus of the tricuspid anterior leaflet, presenting heterogeneous echogenicity and associated with mild valve regurgitation (Figure 1A). Its dimensions, measured by multiplanar 3D echocardiography (Figure 1B) were 2 cm × 1.5 cm. Considering her clinical history, three main differential diagnoses were examined for the intramyocardial mass: a tumor such as papillary fibroelastoma or myxoma, infectious endocarditis, nonbacterial thrombotic endocarditis (NTBE), and thrombus. Urgent cardiac magnetic resonance (CMR) with gadolinium contrast examination was then required. Cardiovascular magnetic resonance (CMR) was performed using a 1.5 Tesla whole-body scanner (Philips Achieva; Phillips Healthcare, The Netherlands). Left and right ventricular volumes and function were within normal ranges. Cine SSFP images confirmed the presence of a large, highly mobile mass with irregular borders in the right ventricle in-flow, probably attached to the tricuspid valve chordae (Figure 2A). This mass appeared to have a low signal intensity, compared to the myocardium, in both STIR T2-weighted and FSE T1-weighted images (Figure 2B,C); however, due to the irregular shape, a contrast agent was administrated to exclude malignancy, and the mass showed no contrast uptake (Figure 2D). Therefore, CMR findings were in keeping with a diagnosis of a large right-ventricular thrombus. The patient was admitted to the Cardiac Intensive Care Unit of University Hospital Federico II, and since the diagnosis of intracardiac thrombus was consistent, anti-coagulant therapy with low-molecular-weight heparin (LMWH) (Enoxaparin sodium at a dose of 6000 IU every 12 h) and acetylsalicylic acid 100 mg, one tablet a day, was started. Although MRI specificity for suspected thrombus is reported to be as high as 99–100% [11], blood culture was carried out to exclude infectious endocarditis. Laboratory tests yielded the following: normal complete blood count, normal electrolytes, and normal creatinine. Partial thromboplastin time was prolonged at 63.6 s and C reactive protein was high at 26.10 mg/L (normal range: 0–5.0 mg/l). Taking into account the occurrence of intracardiac thrombus, the patient’s previous history of pleuropericarditis, pregnancy morbidity, and altered laboratory values, a thrombophilic screening was required to rule out antiphospholipid syndrome (APS). Anticardiolipin antibody IgG and anti-b2 glycoprotein-1 IgG tested positive and they were respectively 63 U/mL (cut off value < 20 U/mL) and 200 U/mL (cut off value < 20 U/mL), while lupus anticoagulant was negative, thus the patient received a diagnosis of primary antiphospholipid syndrome, pending confirmation of the laboratory results at 12 weeks. Despite adequate anticoagulant therapy for three weeks, guided by the anti-factor Xa assays for dose adjustment, transthoracic three-dimensional echocardiography showed the persistence of the vegetation attached to the tricuspid valve, without changes in its size. Therefore, a cardiac surgery consultancy was requested. Considering the high risk of thrombo-embolic and mass detachment [12], with subsequent massive pulmonary embolism, an early, lifesaving surgical thrombectomy was recommended. After multidisciplinary counseling with obstetricians, cardiologists, anesthesiologists, and cardiac surgeons, the patient decided to refuse the termination of pregnancy and undergo cardiac surgery while pregnant. The thrombolytic approach was discarded because of the potential inefficacy of this treatment, as reported in a clinical case series concerning giant right atrial thrombi [13]. The surgical procedure regarding right intracardiac thrombus removal has been previously described in detail [14]. After the surgery, a transabdominal ultrasound evaluation confirmed the presence of a viable fetus with a normal heart rate and a normally inserted placenta. The mass was sent for histological examination and was first addressed as a thrombus. Due to the peculiarity of the patient’s history and the unusual location and size of the mass, we asked the pathologists to review the histological slides providing a second opinion; after this second revision, the presence of nonbacterial thrombotic endocarditis (i.e., Libman–Sacks endocarditis) was stated (Figure 3). After two days in the ICU and 7 days in the cardiology department, the patient was discharged with the following home therapy: acetylsalicylic acid 100 mg/die, enoxaparin 6000 UI/die, and hydroxychloroquine sulfate (200 mg every 12 h). Transthoracic echocardiograms performed monthly after discharge revealed preserved bi-ventricular function with complete competence of the tricuspid valve and the absence of additional masses. 

After surgery, a strict obstetrical follow-up was scheduled. The patient underwent an anomaly scan soon after discharge, demonstrating a normally developed female fetus, with normal growth and normal amniotic fluid. Uterine arteries Doppler velocimetry was abnormal, with a mean Pulsatility Index (PI) above the 95th centile and bilateral notch, suggesting an increased risk for preeclampsia and fetal growth restriction. Weekly obstetric scans were performed to check fetal wellbeing, fetal growth, and maternal and fetal Doppler velocimetry. Fetal heart rate was checked to exclude the possibility of congenital heart block as a consequence of anti-SSA positivity. At the 28th gestational week, the patient was diagnosed with deep vein thrombosis of the right inferior limb, despite prophylactic LMWH therapy and antiplatelets. Anticoagulation treatment was then increased to enoxaparin sodium 6000 IU twice a day. The pregnancy was complicated by fetal growth restriction. At 28 weeks of gestation, fetal weight was below the 5th centile, with normal amniotic fluid and normal fetal Doppler velocimetry. At 32 weeks of gestation, a PI of the umbilical artery (UA) above the 95th centile was found, with normal Doppler velocimetry of the ductus venosus (DV) and reduced PI in the middle cerebral artery (MCA) (brain sparing). Upon admission to the hospital, she underwent daily computerized cardiotocography (c-CTG). Fetal Doppler velocimetry in the UA and MCA remained stable until 34 weeks. At 34 weeks, the patient developed gestational hypertension that was controlled with nifedipine (20 mg every 8 h). At 34 + 0 weeks of gestation, antenatal corticosteroid treatment for the prevention of respiratory distress syndrome was performed with two doses of betamethasone (12 mg every 24 h), according to local protocols. At 34 + 3 weeks of gestation, c-CTG presented repeated deceleration and abnormal short-term variability (below 3 ms). Therefore, an emergency cesarean section was performed. A female newborn weighing 1290 g was born. The Apgar score was 8 at 1 min and 10 at 5 min. According to umbilical arterial emogasanalysis, pH was 7.23 and BE-5. The newborn was admitted to the Neonatal Intensive Care Unit and did not require ventilatory support. Already after a week, she reached the weight of 1500 g and started oral feeding. She has been discharged after two months.

## 3. Discussion

We presented a case of a pregnant woman presenting with a large right intraventricular Libman–Sacks endocarditis, in the context of APS, that was surgically removed. She delivered at 35 weeks of gestation and her pregnancy was complicated by preeclampsia and fetal growth restriction. This case raises several points of discussion. Firstly, our patient was in her sixth pregnancy and she had an obstetric history complicated by previous stillbirths and fetal growth restrictions, but the reasons for such complications have not been investigated before. There is actually no consensus about the possibility of offering a thrombophilia screening in the case of obstetric complications such as stillbirths, intrauterine demise, preeclampsia, and fetal growth restriction [10]. However, our case supports the importance of this screening in order to prevent maternal life-threatening complications in subsequent pregnancies or later in life. The occurrence of Libman–Sacks endocarditis is infrequent and usually develops in the context of advanced malignancies and/or autoimmune diseases, such as APS. It is characterized by the deposition of sterile fibrinous and platelet aggregates that develop on the endocardial surface [15]. Vegetations are mostly left-sided, with the mitral valve more frequently involved than the aortic valve [16]. Isolated tricuspid valve Libman–Sacks endocarditis is quite rare [16,17,18,19,20,21,22], thus the surgical treatment of this condition is anecdotal, especially in the absence of concomitant significant valve impairment [23,24,25,26,27,28,29]. This is likely because the Libman–Sacks vegetations are generally small (<0.5 cm) and therefore do not require surgical excision. Current therapeutic guidelines for APS do suggest long-term anticoagulation to prevent thromboembolic events, although high-quality evidence is limited [30]. Since the treatment of right intracardiac mass is based on anecdotical experience [12], all therapeutic options were taken into account, including conservative therapy with anticoagulants, thrombolysis, or surgical excision. Adjusted-dose enoxaparin, administered twice a day for over 2 weeks, did not give appreciable results. A low-dose, slow infusion of tissue-type plasminogen activator appeared to be efficacious in pregnant women [31]; however, the stillbirth rate is about 20%. Indeed, though recombinant tissue plasminogen activator does not cross the placenta, it may induce bleeding complications, such as sub-placental bleeding. The risk evaluation of performing an early cardiac surgery compared to a delayed (at the end of pregnancy) approach needs to address both maternal and fetal perioperative risk and thromboembolic risk. The heterogeneity of clinical conditions leading to cardiac surgery during pregnancy limits the predictability of maternal and fetal outcomes during cardiopulmonary bypass [32]. Risk factors for maternal mortality during cardiac surgery include the use of vasoactive drugs, age, type of surgery, reoperation, and maternal functional class [33]. Risk factors for fetal mortality include maternal age >35 years, functional class, reoperation, emergency surgery, type of myocardial protection, and anoxic time [33]. A recent review reported that the mortality rate after cardiac surgery during pregnancy is around 10% and 30% for the mother and the fetus, respectively [34]. Therefore, it has been commonly suggested to postpone cardiac surgery after the delivery, when possible. On the other hand, in patients affected by nonbacterial thrombotic endocarditis, the occurrence of pulmonary emboli has been reported to be higher than 50% [35]. In fact, unlike infective endocarditis, clinical manifestations of nonbacterial thrombotic endocarditis are deceitful and result more frequently in systemic emboli rather than valvular dysfunction. Finally, once agreed on the presumed high embolic risk, our multidisciplinary team considered the maternal and fetal perioperative mortality rates to be acceptable and decided to perform the surgical excision of the vegetation. Adverse pregnancy events are very frequent in women with APS, even if very recent published data says that 65% of high referral APS pregnancy cohorts resulted in term live delivery [36]. The management of patients who are aPL carriers has not been standardized yet, but there is a general consensus that the patients who do not have thrombocytopenia, thromboembolic phenomena, or pregnancy morbidity should not be submitted to any treatment or should only take low doses of acetylsalicylic acid. In the presence of any one of the conditions above, various treatment regimens have been used (corticosteroids, dipyridamole, heparin, warfarin, cytotoxic drugs, plasmapheresis, high-dose human immunoglobulin) according to the severity and individuality of the case. In this case, due to the contraindication for the use of oral anticoagulants in a pregnant woman, we used heparin during pregnancy in association with aspirin and hydroxychloroquine. In conclusion, APS should be strongly suspected in any patient with echocardiographic evidence of valvular thickening or valve nodules and a history of pregnancy losses and/or thromboses. The indication and timing of the surgical intervention must be decided by a multidisciplinary team, cautiously addressing the risks and benefits of an early compared to a delayed approach.

## 4. Conclusions

Intracardiac thrombosis or non-bacterial thrombotic endocarditis is a rare complication of antiphospholipid syndrome. Evidence relating to Libman–Sacks endocarditis cases in pregnant women with APS is still very limited, so we think that reporting this case might help in clinical practice. The correct management of pregnancy in patients with this syndrome is fundamental: a cardiological visit with a multidisciplinary approach should be required. The best management and the most appropriate treatment are essential for a successful pregnancy.

## Figures and Tables

**Figure 1 jcm-11-05875-f001:**
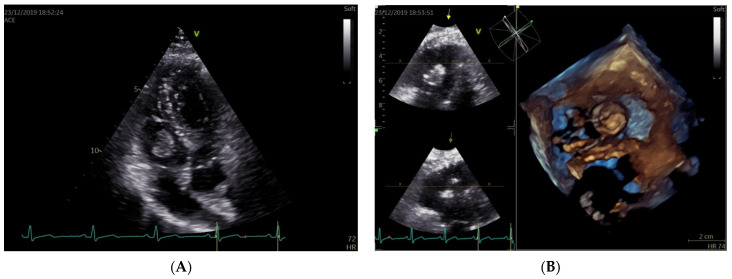
Standard (**A**) and 3D echocardiographic assessment (**B**) of right-side intracardiac mass.

**Figure 2 jcm-11-05875-f002:**
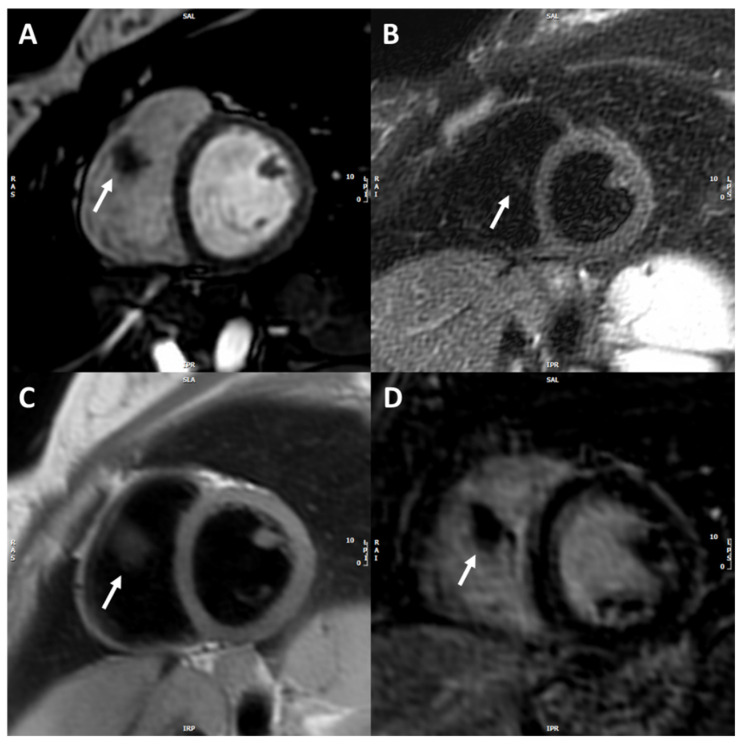
Cardiac magnetic resonance depicting right non-vascularized intracardiac mass. CMR images: Cine SSFP short-axis image (**A**) showing the mass in the right ventricle (white arrow) with irregular borders. STIR T2-weighted (**B**) and FSE T1-weighted short-axis images (**C**) showing the lower signal intensity of the mass compared to the myocardium. Late post-contrast short-axis image (**D**) showing no-contrast uptake from the mass.

**Figure 3 jcm-11-05875-f003:**
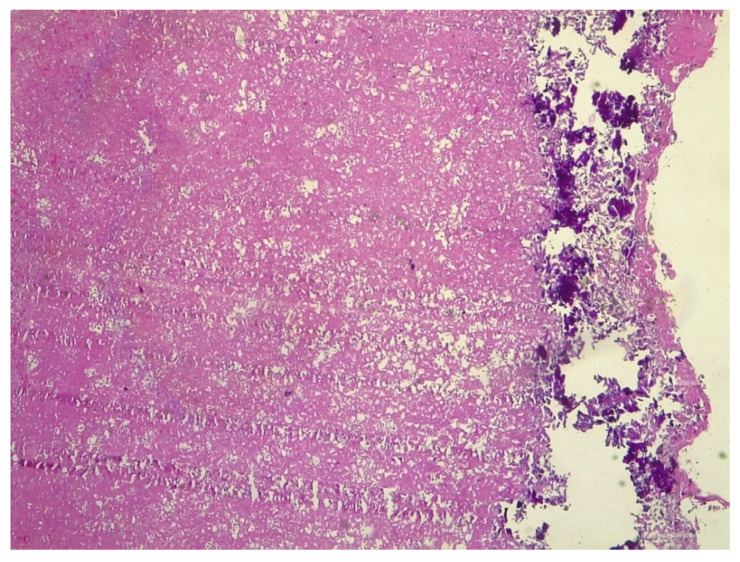
Histological specimen of the intracardiac mass revealing nonbacterial thrombotic endocarditis (i.e., Libman–Sacks endocarditis).

## Data Availability

Not applicable.

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
