# Peer review of "A Rare Case of Tricuspid Valve Libman–Sacks Endocarditis in a Pregnant Woman with Primary Antiphospholipid Syndrome"

_jcm, 2022, doi:10.3390/jcm11195875_

Round 1

Reviewer 1 Report

Dear Sonia Migliorini and coauthors with great interest, I read your article entitled "A rare case of tricuspid valve Libman-Sacks endocarditis in a pregnant woman with primary antiphospholipid syndrome." I think the topic is interesting for scientific and medical audiences. The article is very well written and clearly structured. The abstract is informative, the clinical data of the patient are sufficiently described, and the conclusion is informative. I have only thoughts about the introduction. The authors could give a brief description of Libman-Sacks endocarditis (LSE) in the introduction. There are also reports of Libman-Sacks endocarditis (LSE) in patients with SLE and also in patients with APS. However, there is no report of LSE in pregnant women with APS. This is something that the authors might want to highlight to emphasize the importance of their case report.

I have only a few minor comments:

-     -   Abstract: Line 42, acid. acetylsalicylic should be replaced with acetylsalicylic acid.

-      -    Introduction: On page 2 line 66, there is an additional period at the end of a sentence before the reference that should be omitted.

-      -    Page 67: lupus erythematosus, replace with Systemic lupus erythematosus

-     -     Case Report: page 2 line 79: ''She was at her sixth pregnancy'' contradicts the discussion: page 6, line 9: '' Patient was at her fifth pregnancy'' What is correct?

-    -      Page 7, line 240, pregnancy, probably you meant pregnancy morbidity

-   -       Page 7: line 246, antiphospholipid syndrome should be replaced by APS.

Reviewer 2 Report

This paper - a case report shows intracardiac thrombosis or non-bacterial thrombotic endocarditis as a rare complication of Antiphospholipid Syndrome in pregnancy. The paper and the discussion state a multidisciplinary approach to the mentioned rare clinical problem. The paper was written according to the journals propositions, extremely clinically attractive and instructive with a description of the modalities of approach and treatment of complex syndrome in pregnancy. 

Author Response

I thank the reviewer for his words. The multidisciplinary approach was essential given the rarity of the clinical problem to offer the patient the best treatment available to us.

Reviewer 3 Report

Although rare in clinical occurrence, a significant malady with serious cardiac consequences for a patient (and a fetus in this case, as well) is well described in this case report. The course of diagnostical work-up, clinical considerations and course-adjusted successful clinical management are well presented. Well-written discussion gives valuable hints for approach to women with history of failed pregnancies.

Author Response

I thank the reviewer for his words. The problem that the patient had to face, gave us an opportunity to question ourselves in approaching all pregnancies